# Tensile Microstrain Fluctuations in the BaPbO Units in Superconducting BaPb$_{1-x}$Bi$_x$O$_3$ by Scanning Dispersive Micro-XANES

Ruben Albertini [1], Salvatore Macis [1,2,*], Andrei A. Ivanov [3], Alexey P. Menushenkov [3], Alessandro Puri [4], Virginia Monteseguro [5], Boby Joseph [6], Wei Xu [1,7], Augusto Marcelli [1,8], Paula Giraldo-Gallo [9], Ian Randal Fisher [10,11,12], Antonio Bianconi [1,13,*] and Gaetano Campi [1,13,*]

[1] RICMASS Rome International Centre Materials Science Superstripes Via dei Sabelli 119A, 00185 Rome, Italy; rubalbertini@gmail.com (R.A.); xuw@ihep.ac.cn (W.X.); augusto.marcelli@lnf.infn.it (A.M.)
[2] Department of Physics, Sapienza University Rome, P.le Aldo Moro 2, 00185 Rome, Italy
[3] Department of Solid State Physics and Nanostructures, National Research Nuclear University MEPhI (Moscow Engineering Physics Institute), 115409 Moscow, Russia; andrej.ivanov@gmail.com (A.A.I.); apmenushenkov@mephi.ru (A.P.M.)
[4] CNR-IOM-OGG c/o ESRF-The European Synchrotron, 71 Avenue des Martyrs, 38000 Grenoble, France; alessandro.puri@esrf.fr
[5] Condensed Matter Physics Department, MALTA Consolider Team, Facultad de Ciencias, University of Cantabria, 39005 Santander, Spain; virginia.monteseguro@unican.es
[6] Elettra-Sincrotrone Trieste, Basovizza, 34149 Trieste, Italy; boby.joseph@gmail.com
[7] Institute of High Energy Physics (IHEP), Chinese Academy of Sciences, Beijing 100039, China
[8] INFN—Laboratori Nazionali di Frascati, 00044 Frascati, Italy
[9] Department of Physics, Universidad de Los Andes, Bogotá 111711, Colombia; pl.giraldo@uniandes.edu.co or pgiraldog@gmail.com
[10] Department of Applied Physics, Stanford University, Stanford, CA 94305, USA; irfisher@stanford.edu
[11] Geballe Laboratory for Advanced Materials, Stanford University, Stanford, CA 94305, USA
[12] Stanford Institute for Materials and Energy Sciences, SLAC National Accelerator Laboratory, 2575 Sand Hill Road, Menlo Park, CA 94025, USA
[13] Institute of Crystallography, National Research Council of Italy, Via Salaria Km 29.300, Monterotondo Stazione, 00015 Rome, Italy
* Correspondence: salvatore.macis@uniroma1.it (S.M.); antonio.bianconi@ricmass.eu (A.B.); gaetano.campi@ic.cnr.it (G.C.)

**Abstract:** BaPb$_{1-x}$Bi$_x$O$_3$ (BPBO) bismuthate, showing high T$_C$ superconductivity for 0.05 < x < 0.35, is an archetypal system for studying the complex inhomogeneity of perovskite lattice favoring the emergence of quantum coherence, called the superstripes phase. Local lattice fluctuations, detected by EXAFS; nanoscale stripes, detected by electron microscopy; and two competing crystalline structures, detected by diffraction, are known to characterize the superconducting phase. At nanoscale [BaBiO$_3$] centered nanoscale units (BBO) coexist with BaPbO$_3$ centered (BPO) units in the BPBO perovskite; therefore, we expect a tensile microstrain in BPO units due the misfit strain between the two different lattices. Here, we report the measurement of the spatial micro-fluctuations of the local tensile microstrain $\varepsilon$ in the BaPO units in superconducting Ba(Pb$_{1-x}$Bi$_x$)O$_3$ crystals with x$_1$ = 0.19 an x$_2$ = 0.28. We show here the feasibility of applying the scanning dispersive micro-X-ray absorption near edge structure (SdμXANES) technique, using focused synchrotron radiation, to probe the microscale spatial fluctuations of the microstrain in BPO units. This unconventional real-space SdμXANES microscopy at the Pb $L_3$ edge has been collected in the dispersive mode. Our experimental method allows us to measure either the local Bi chemical concentration x and the local lattice microstrain of local BBO and BPO units. The 5 × 5 micron-size spots from the focused X-ray beam allowed us to obtain maps of 1600 points covering an area of 200 × 200 microns. The mapping shows a substantial difference between the spatial fluctuations of the microstrain $\varepsilon$ and the chemical inhomogeneity x. Moreover, we show the different relations $\varepsilon(x)$ in samples with lower (x$_1$ = 0.19) and higher (x$_2$ = 0.28) doping respect to the optimum doping (x = 0.25).

**Keywords:** scanning dispersive micro XANES; SdµXANES; $BaPb_{1-x}Bi_xO_3$; microstrain; high-Tc superconductivity; strain; $BaPbO_3$ units; $BaBiO_3$ units

## 1. Introduction

The critical temperature of the superconducting transition and electronic structure at the Fermi level in quantum complex matter and high temperature superconductors is governed not only by doping, but also by the lattice strain in nickelates, iron-based superconductors, and heterostructures [1–24] controlling the complex landscape generated by nanoscale phase separation [24–29] near the superconductor–insulator transition [30], as it was first shown in cuprates and diborides [18–23]. In this work we apply scanning µ-X-ray absorption near edge structure (SµXANES) to unveil the spatial heterogeneity of local lattice fluctuations [31–35] in a selected nanocluster related to the strain at specific sites, namely, microstrain spatial fluctuations near the metal to insulator transition in $BaPb_{1-x}Bi_xO_3$. [36–39].

The dimorphic crystal structure of $BaPb_{1-x}Bi_xO_3$ superconducting compounds is a distorted perovskite ($ABO_3$) for the highest Bi concentrations and contains two Bi sites with different Bi-O bond lengths [40–52]. $BaPb_{1-x}Bi_xO_3$ can be described as a composite heterostructure made of p-n junctions formed by a low charge carrier density ($10^{21}$ $cm^{-3}$) metallic BaPbO nanoscale units, and a charge density wave (CDW) of semiconducting BaBiO nanoscale units. With increasing Pb content, the insulating CDW phase disappears, giving rise to a metallic phase, where superconductivity appears at x > 0.05 and the metal insulator transition at x = 0.35. Recently, there has been increased interest in the artificial synthesis of thin film heterostructures providing the signature of 2D interface superconductivity in a landscape of nanoscale phase separation [53–60]. X-ray absorption near edge structure, called XANES spectroscopy, [61–68] provides a local and fast probe of the local lattice fluctuations probing nanoscale phase separation. It has been applied to investigate powder samples of $BaPb_{1-x}Bi_xO_3$ [69–71] but its spatial inhomogeneities are unknown.

The $BaPb_{1-x}Bi_xO_3$ structure [36–39], shown in Figure 1a, has a rich phase diagram due to the inter-relation between multiple perovskite structures as a function of temperature and doping [72–75]. In the region of 0.05 < x < 0.35, the diamagnetic signal increases and the material becomes superconducting. In this regime, its structure is polymorphic [36–39] with a fraction of its volume with orthorhombic *Ibmm* symmetry and the rest with tetragonal *I4/mcm* symmetry. The critical temperature $T_C$ as a function of the bismuth content x for the samples under study is presented in Figure 1b. For x values over the superconducting range until about x = 0.77, only the orthorhombic *Ibmm* symmetry is observed. In the $BaBiO_3$ phase, the alternation of $Bi^{3+}$ and $Bi^{5+}$ leads to structural distortions involving both the so-called breathing distortions and the tilting of $BiO_6$ octahedra, which lowers the crystal symmetry from a cubic to a monoclinic space group.

High-resolution transmission electron microscopy (HRTEM) has been used to unveil the emergence of structural stripes at the nanoscale in single crystals of $BaPb_{1-x}Bi_xO_3$ [39]. The stripe-like morphology found in this study has a deep significance; it might provide a natural clue to understanding the characteristic behavior observed at the superconductor–insulator transition close to optimal doping in $BaPb_{1-x}Bi_xO_3$, motivating a theoretical investigation of the percolation effects near the quantum phase transition in any material with a striped morphology.

While it is known that a misfit strain generates a complex nanoscale phase separation in cuprates [19–28] and iron-based superconductors with a percolative network, the misfit strain in $Ba(Pb_{1-x}Bi_x)O_3$ has not yet been reported. In this work, we have used SµXANES to investigate the spatial distribution of microstrain in three $BaPb_{1-x}Bi_xO_3$ samples with x = 0, 0.19 and 0.28 (full red circles in Figure 1b) to obtain a deeper insight into the relationship between local lattice inhomogeneity and superconductivity in this system.

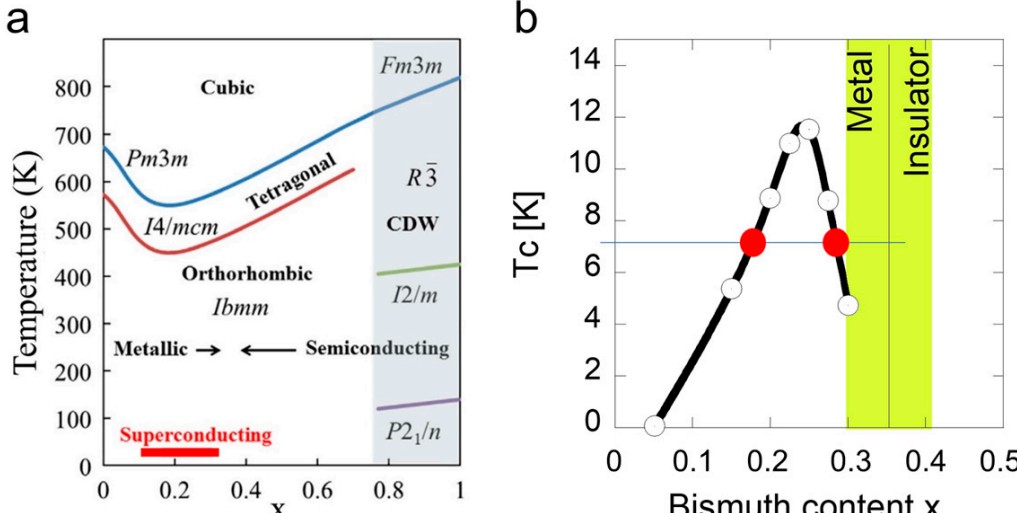

**Figure 1.** (**a**) Phase diagram for the BaPb$_{1-x}$Bi$_x$O$_3$ with the different space groups indicated. The CDW region is indicated with shaded colors, while the superconducting range 0.05 < x < 0.35 is outlined by the red bar. (**b**) Superconducting critical temperature T$_c$ as a function of the bismuth content, x. The white dots from Climent-Pasqual et al. [36] and the full red circles represent the nominal concentrations of the doped BaPb$_{1-x}$Bi$_x$O$_3$ samples investigated. The green region indicates the metal insulator transition centered at x = 0.36.

## 2. Results

### 2.1. Variation of the Local Structure of BaPbO$_3$ 0.5 nm Nano Cluster Units in BaPb$_{1-x}$Bi$_x$O$_3$

In intrinsically inhomogeneous composites, in nanoscale units, the microstrain is generated by the lattice misfit strain originating from the coexistence of different units with different lattice parameters occurring during the cubic-to-tetragonal and tetragonal-to-orthorhombic transitions. This mechanism results in a multi-domain microstructure with many incoherent boundaries, and originates the local lattice microstrain $\varepsilon$.

It has been proposed that the mobility of the charge carriers, moving through octahedra, may be favored by the tetragonal symmetry while carriers' localization takes place when the symmetry decreases to the orthorhombic and then to the monoclinic phase with an increasing bismuth content; still, the spatial inhomogeneity of charge and lattice remains unknown.

### 2.2. Scanning Micro XANES Microscopy: Microstrain Maps and Phase Separation

The SµXANES technique is a unique tool to investigate the local atomic lattice structure via multiple scattering resonances of the excited photoelectron probing the geometry of the local clusters surrounding selected atomic species. The local structure of the atomic cluster centered at the Pb-absorbing atom is shown in Figure 2a.

We have first calculated the XANES profiles at the Pb $L_3$ edge for the cluster with the Pb ion at the center and different microstrain values using the full multiple-scattering approach. Calculated Pb $L_3$ XANES are shown in Figure 2b, while the relative variation of the energy of the highest peak due to the multiple scattering resonance (MSR) as a function of the tensile or compressive microstrain is presented in Figure 2c.

The spatial distribution of the strain at different doping has been investigated by SµXANES using focused synchrotron radiation that allowed us to identify the spatial inhomogeneity at the micron scale related to incipient phase separation. Samples of superconducting BaPb$_{1-x}$Bi$_x$O$_3$ single crystals with x = 0, 0.19 and 0.28 were scanned on a 200 × 200 µm$^2$ area with a 5 µm step in both directions. Data have been collected at the ESRF beamline ID24 (Grenoble, France), which is equipped with an energy dispersive X-ray spectrometer and a unique setup for real-space scanning with variable temperature [7].

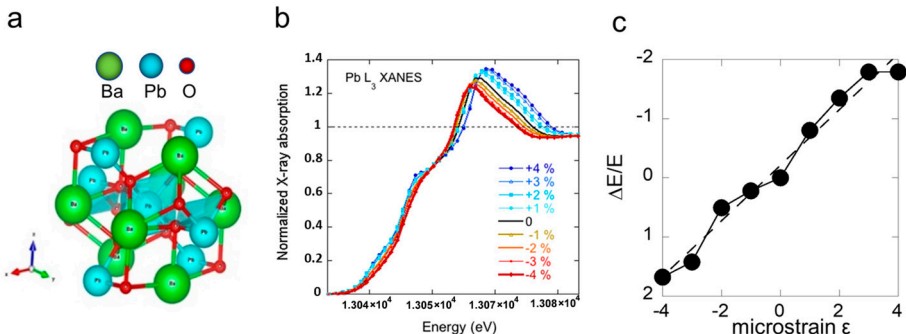

**Figure 2.** Panel (**a**) The nanoscale cluster of 29 atoms centered at the Pb-absorbing atom called the BPO unit where the photoelectron is confined at the multiple scattering resonance (MSR) giving the main peak in the Pb $L_3$ XANES shown in panel (**b**). Panel (**b**) Full multiple scattering Pb $L_3$ edge XANES spectra for the BPO cluster in panel (**a**) calculated by increasing the lattice parameter of the 0.5 nm nano cluster, shown in panel (**a**), relative to the original lattice parameter in the nanocluster in undoped BaPbO$_3$, which called XANES microstrain values for the nano BaPbO units. Panel (**c**) shows the relative variation of the energy of the highest peak in the XANES spectrum due to the multiple scattering resonance (MSR) ΔE/E relative to the first weak absorption peak in the edge region as a function of the microstrain of the lattice parameter in the nano 0.5 nm cluster shown in panel (**a**) used in the XANES calculation using the FEFF9.6 code [68], the dashed line is a linear fit.

In Figure 3, we show XANES spectra at the Pb $L_3$ edge, where the maximum absorption peak is due to the multiple scattering resonance (MSR) of the excited photoelectron, also called the Fano shape resonance in the continuum, at about 32 eV above the absorption threshold taken as zero energy. Alongside XANES spectra, we also show their first derivatives where we clearly recognize three peaks. The energy at 13,027 eV has been chosen as the reference zero value for the energy scale while the energy of the multiple scattering resonance ($E_{MSR}$) has been defined as the difference between the zero of the derivatives, $E_0$, and the first derivative peak, $P_1$:

$$E_{MSR} = E_0 - P_1 \tag{1}$$

MSR energy, $E_{MSR}$, of the photoelectron confined in a nanoscale BPO cluster centered at the Pb-absorbing atom probes the variations of the average Pb–O bond length and its spatial variation. In this regard, $E_{MSR}$ is a sensitive measure of the local lattice strain at Pb sites. It is fundamental to remember that slight variations of the average bond distance (R) can be attributed to the dependence of phase shifts in the framework of the multiple scattering model [39].

It is known that the energy variation of the main multiple-scattering resonance in the XANES spectra is connected with Pb–O average distance in the cluster, $R$, by

$$E_{MSR} \propto R^{-2} \tag{2}$$

where the XANES microstrain can be calculated as

$$\frac{\Delta R}{R} = -\frac{1}{2} \frac{\Delta E_{MSR}}{E_{MSR}} \tag{3}$$

where

$$\Delta E_{MSR} = E_{MSR} - \overline{E_{MSR}} \tag{4}$$

In the undoped (x = 0) BaPbO$_3$ sample, the peak around its average value $\overline{E_{MSR}}$ is relatively narrow and has been taken as reference for measuring the microstrain variation.

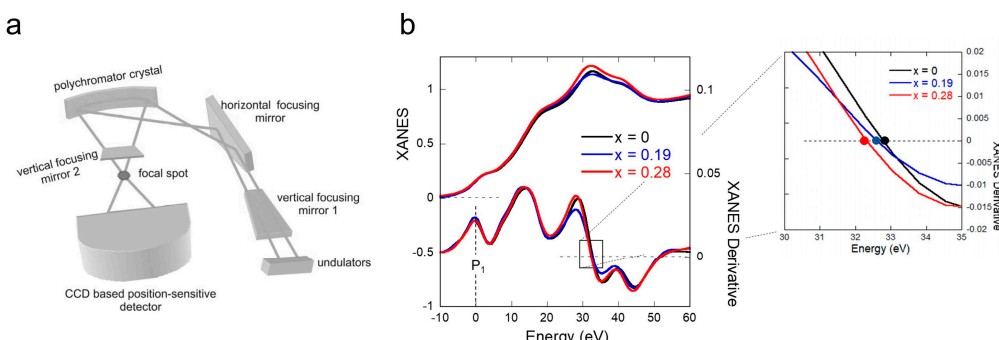

**Figure 3.** Upper panel (**a**) The experimental set up to collect X-ray absorption near edge structure (XANES) in the dispersive mode at the ID24 ESRF beamline. The X-ray synchrotron radiation emitted by the undulator is focused on a 5 μm spot on the thin crystal sample. The X-ray absorption is measured in transmission, and we recorded both the spatial variation of the bismuth atomic concentration x in superconducting crystals of $BaPb_{1-x}Bi_xO_3$ and the spatial fluctuations of the lattice strain in selected $BaPbO_3$ nanoclusters. Lower panels (**b**) X-ray absorption near edge structure (XANES) at the Pb $L_3$ edge of $BaPb_{1-x}Bi_xO_3$ samples with x = 0, 0.19 and 0.28, where the maximum absorption peak is due to the multiple scattering resonance (MSR) of the excited photoelectron in the continuum above the absorption threshold taken as zero energy. The derivatives of the signal from Pb show three peaks. $E_0$ at 13,027 eV has been chosen as reference zero value for the energy scale. The energy of the multiple scattering resonance ($E_{MSR}$) has been defined as the difference between the zero of the first derivative peak (P1). The energy of the MSR of the photoelectron confined in a nanoscale cluster, is highlighted by full circles in the rectangular inset of the derivative.

Measurements of single crystals of $BaPb_{1-x}Bi_xO_3$ with different Bi compositions around the superconducting maximum [37] (referring to the diagram 1a) show signatures of granular and inhomogeneous superconductivity possibly associated with a nanoscale structural phase separation. Due to different values of Bi valence and the variation of Pb concentration, $BaPb_{1-x}Bi_xO_3$ is polymorphic where different structural phases coexist. It is interesting to analyze how these phases appear in the lattice.

From Pb $L_3$-edge XANES spectra, we obtained real space maps of the $E_{MSR}$ for different doping of $BaPb_{1-x}Bi_xO_3$; then, we have calculated the microstrain referring to Equation (3). Here, we report the micro XANES data results for x = 0.19 and x = 0.28 crystalline samples collected at low temperatures of 50 ± 10 K. The spatial fluctuations of the slight energy shift of the multiple scattering resonance energy in the superconducting Bi-doped samples are compared to the data for the reference undoped $BaPbO_3$ sample. The decreasing trend of the negative energy shift of the main multiple scattering resonance $E_{MSR}$ is attributed to a lattice expansion [39] of the Pb-centered nanoscale puddles increasing with the negative XANES tensile strain.

Figure 4a shows the microstrain maps at different Pb contents. Between the insulating (x = 0) and metallic (x = 0.28) phases, the sample with x = 0.19 presents a large inhomogeneity with a phase separation where microstrain assumes values of insulating and metallic phases in different spatial regions. This disorder behavior is measured by the probability distribution function (PDF), shown in Figure 4b, from which information about lattice strains can be obtained. Indeed, comparing the $E_{MSR}$ value of each point with the average value of the undoped sample, we obtained the local lattice strain for our differently doped compounds.

The extended X-ray absorption spectra of the $BaPb_{1-x}Bi_xO_3$ at either the Pb $L_3$ jump and the Bi $L_3$ edge jump in the stoichiometric compounds at x = 0 and x = 1 allow the direct measure of the bismuth content x at each spot where the microstrain is measured (see Figure 5a).

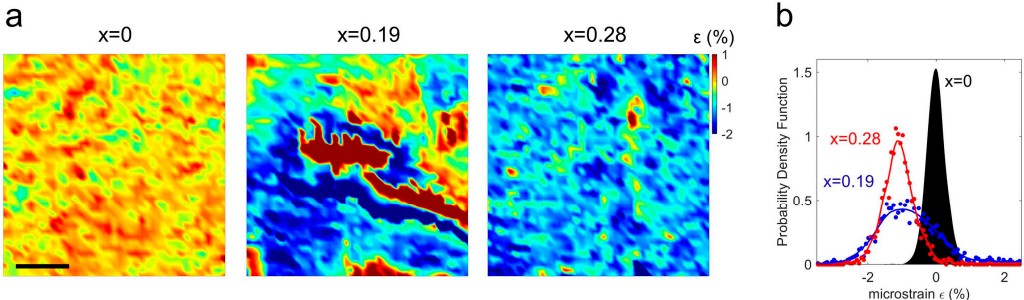

**Figure 4.** Panel (**a**) show the maps of spatial fluctuations of the *microstrain* over a square of $200 \times 200$ microns measured by $\Delta R/R = -\Delta E_{MSR}/2E_{MSR}$ for the undoped (BaPbO$_3$), x = 0, and the BaPb$_{1-x}$Bi$_x$O$_3$ doped samples with x = 0.19 and x = 0.28. The bar corresponds to 25 $\mu$m. Panel (**b**) shows the probability distribution function (PDF) of the *microstrain* extracted from the three BaPb$_{1-x}$Bi$_x$O$_3$ maps shown in panel (**a**). For the undoped non-superconducting sample, with x = 0, the distribution's dark PDF peak is centered at zero microstrain, the distribution of the microstrain in the superconducting sample with x = 0.28 (red curve) is centered at around 1% with no overlap with the PDF distribution of the undoped reference sample. The distribution of the microstrain in the x = 0.19 sample (blue dots) shows a broad distribution overlapping the non-superconducting phase, and the spatial map clearly shows microscale phase separation.

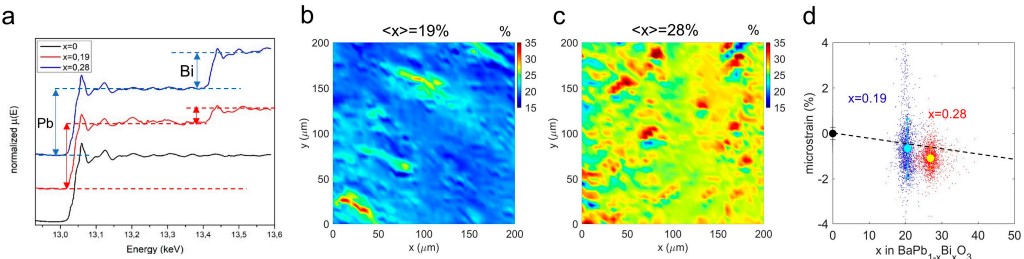

**Figure 5.** (**a**) XAS spectrum of the BaPb$_{1-x}$Bi$_x$O$_3$ over a large range including either the Pb $L_3$ edge and the Bi $L_3$ edge jumps plotted with spectra of stoichiometric compounds at x = 0 and x = 1. The ratio between the amplitude of the absorption jumps of both Bi and Pb edges allows the direct measurement of the bismuth content, x, at the same spot where the microstrain is measured. Panels (**b**,**c**) show the maps of Bi x content measured by the absorption jump ratio of the XANES profile going from Pb $L_3$ to the Bi $L_3$ edge. (**d**) The local macrostrain plotted vs. the local Bi content x in two BaPb$_{1-x}$Bi$_x$O$_3$ samples with nominal x = 0.19 (blue squares) and x = 0.28 (red circles). The full blue and yellow full circles represent the mean values of tensile local XANES microstrain [61–64] in the x = 0.19 and x = 0.28 samples, respectively. The dashed line indicates the average lattice *strain* as a function of Bi content, measured by diffraction [36], which agrees with the Vegard's law. Very large fluctuation of local XANES microstrain are unveiled at microscale by our experimental method.

In this way, we have obtained the maps of *x* content in the two samples with nominal content x = 0.19 and x = 0.28 shown Figure 5b,c, respectively.

In Figure 5d we visualize the microstrain in the BaPb$_{1-x}$Bi$_x$O$_3$-doped samples vs. the Bi content x, where the x = 0.19 and x = 0.28 samples are represented with blue squares and red circles, respectively. The full circles represent the mean values of microstrain and x content in the x = 0.19 x = 0.28 samples, respectively.

The lattice tensile strain as a function of Bi content x measured by crystallography lattice parameters in BaPb$_{1-x}$Bi$_x$O$_3$ going from x = 0 to x = 1 follows Vegard's law represented by the dashed line $\varepsilon = -2x$ [36] and the spatial average local microstrain in the Pb-centered clusters in the two BaPb$_{1-x}$Bi$_x$O$_3$ samples is indicated by blue and yellow dots in Figure 5. In both samples, the average local tensile strain in Pb-centered clusters is larger than that measured in the average lattice crystal indicated by Vegard's law.

### 3. Discussion

We reported local measurements of the lattice strain of $BaPb_{1-x}Bi_xO_3$ for different values of doping. In complex perovskite materials, different structural phases are in competition, and their stability is tuned by the microstrain. These phases cause a local change in the atomic strain that can be observed in the MSR of XANES spectra. The $BaPb_{1-x}Bi_xO_3$ superconductor shows a high-$T_C$ superconductivity for $0.05 < x < 0.35$, becoming an archetypal system with competitions of different nanoscale units with local lattice symmetry [36–40] which favor the emergence of macroscopic quantum coherence in complex nanoscale phase separation [36–40]. This compound could be described as the complex solid solution of two nanoscale clusters, resulting from metallic $[BaPbO_3]_{(1-x)}$ and insulating $[BaBiO_3]_{(x)}$ perovskite units, which are expected to be strained due to the misfit between their different lattices. In the underdoped superconducting phase at $x = 0.19$ the Pb-centered nanocluster units probed by Pb $L_3$ XANES spectroscopy shown in Figure 2a called BaPO unit shows large spatial tensile microstrain fluctuations in the range $-0.1\% < \varepsilon < -2\%$, while the Vegard's law expected value is $-0.38\%$.

In the overdoped regime at $x = 0.28$ the tensile microstrain fluctuations are strongly reduced and centered $\varepsilon = -1\%$ which is larger than the expected value of $-0.5\%$ predicted by the Vegard's law as indicated in the figure. Therefore in the overdoped regime the deviation of the local microstrain $\varepsilon$ in the probed [BPO] nanoscale cluster units from the average lattice strain measured by X-ray diffraction [36] indicates a nanoscale phase separation between first metallic units and insulating units, triggering high-temperature superconductivity, which is in agreement with high resolution electron diffraction [40].

### 4. Materials and Methods

Single crystals of $BaPb_{1-x}Bi_xO_3$ with Bi compositions $x = 0$, 0.19 and 0.28 were grown by slowly cooling a solution of $BaCO_3$, PbO, and $Bi_2O_3$. The single crystals were polished in the Geballe Laboratory for Advanced Materials in Stanford to reach the optimal homogeneous hundred microns thickness for X-ray transmission experiments at the Pb $L_3$ edge energy in the high X-ray energy range 13 KeV.

The in-depth comprehension of local structural phases requires tools not accessible in ordinary laboratories due to the need to probe micrometric areas without moving the sample, as well as short acquisition times. Given these non-trivial premises, the micro-XANES technique makes possible the detection of nanoscopic local lattice fluctuations in in-homogeneous systems. The energy dispersive technique, necessary for studying $BaPb_{1-x}Bi_xO_3$ [76], has become available thanks to the dedicated synchrotron radiation beamline ID24 of the European Synchrotron Radiation Facility (ESRF) in Grenoble, France. The ID24, consists of a set of mirrors in a Kirkpatrick–Baez geometry (vertical and horizontal focusing mirrors) that focus the white beam on a polychromator crystal, which splits the beam. A second vertical mirror focuses the beam on the sample. A photon flux of $\sim 10^{14}$ ph/s goes through the sample; the transmitted photons are detected by a Ge strip CCD equipped with an acquisition electronics with a readout time of 2.8 µs and minimum integration time of $\sim$80 ns. The particular setup of the beamline, and the lack of moving parts provide a small and stable focal spot ($5 \times 5$ µm$^2$ at Pb and Bi $L_3$-edges) suitable for the scanning micro-XAS. [7]. The reported XANES data were collected at low temperatures of $50 \pm 10$ K.

In order to establish the actual limitation of the measure of the local cluster microstrain by scanning micro XANES in the dispersive mode, we present in Figure 6, the typical XANES spectra collected in 12 spots of the 1681 XANES spots collected in the area of $41 \times 41$ spots of the pure $BaPbO_3$ sample shown in Figure 4. The energy of the multiple scattering resonance ($E_{MSR}$) is given by the difference between the Fermi level at the maximum of first derivative peak and the zero of the XANES spectra derivative around 32.8 eV. The energy fluctuations of the multiple scattering resonance ($E_{MSR}$) due to the noise level correspond to a noise fluctuation of the microstrain of the order of 0.12% in the

undoped sample. Therefore, we can clearly measure microstrain spatial fluctuations larger than 0.12% in the doped superconducting samples.

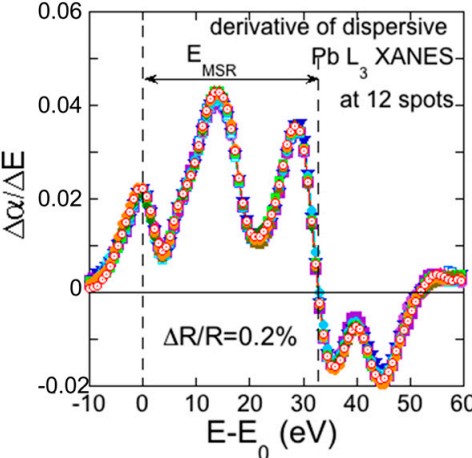

**Figure 6.** Typical 12 derivatives of Pb $L_3$ XANES spectra collected using the dispersive mode in 12 spots of the 1681 spots measured by scanning mode in BaPbO$_3$ sample where the photon energy scale is measured from the absorption edge E$_0$ at 13,027 eV. The different color of the dots indicates different XANES spectra in different spatial.

Theoretical simulations of XANES spectra were performed in the framework of full multiple scattering theory for a finite atomic cluster centered at the absorbing Pb atom using the muffin-tin approximation [61–64] and self-consistent field using the implemented FEFF9.6 code, as described in reference [68]. The Hedin–Lundqvist exchange potential was adopted and the atomic potential was constructed in self-consistent field calculations. The full multiple scattering (MS) calculations were calculated increasing the atomic cluster radius centered at the Pb-absorbing atom. The calculated XANES spectra reproduced the experimental data for clusters with a radius larger than 5 Å, and remained invariant for larger clusters. This result shows that the experimental XANES spectrum probes only a BPO nanoscale cluster of 5 Å radius via multiple scattering. The instrumental and core–hole-induced broadening were consistently used for all simulated spectra to be comparable with experimental spectra.

### 5. Conclusions

We have measured the spatial distribution of microstrain of Pb-centered clusters with two different Bi contents (x) in BaPb$_{1-x}$Bi$_x$O$_3$ using scanning dispersive μ-XANES (Sdμ-XANES). Our measurements reveal a relative variation of the Pb-O bond, as determined from the energy shift of the multiple scattering resonance $E_{MSR}$. We have found that the largest inhomogeneity occurs in the under-doped regime with lower Bi dopant concentrations, with x = 0.19, close to the superconductor-to-metal transition (SMT) for the weak tensile *microstrain.* At variance, the sample with the higher Bi content x = 0.28, near the superconductor-to-insulator (SIT) transition shows a larger tensile strain in the Pb-centered nanoscale clusters probed by Pb $L_3$-XANES, called [BaPbO] nanoscale units Finally, we have shown that the amplitude of the spatial fluctuations of the tensile microstrain in the [BaPbO] nanoscale units is about −0.5% and −0.8% in the x = 0.28 and x = 0.19 samples, respectively. In these two samples, the average tensile microstrain was quite a bit larger than predicted by Vegard's law, i.e., −0.38% and −0.56%. We assign these results to nanoscale phase separation with the first nanoscale units of metallic strained $\varepsilon < 0.2$ and insulating $\varepsilon > 0.72\%$ units which favors interface superconductivity triggered by the nanoscale phase separation. While the two samples investigated in this work have similar critical temperatures, the x = 0.19 sample is made of larger metallic units and a smaller insulating phase and the x = 0.28 sample is made of larger insulator units and

smaller metallic units. While in the literature many authors usually assume that in these systems the lead local structure is not affected by Bi content, we have shown that the local structure near the Pb site is affected by Bi doping. Finally, this work shows the feasibility of spatial mapping intrinsic inhomogeneity by scanning micro XANES in the dispersive mode, which can be implemented by advances on focusing the X-ray beam down to the nanoscale to provide further unique additional information on the micro fluctuations of local complexity at the atomic scale in quantum materials [77,78].

**Author Contributions:** Conceptualization, P.G.-G., S.M. and A.B.; methodology, A.A.I., A.P.M., A.M., W.X., A.B. and G.C.; software, G.C.; validation, A.P.M., A.B. and A.M.; formal analysis, R.A., S.M., G.C. and A.B.; investigation, B.J., A.P., V.M., S.M. and A.A.I.; sample synthesis, P.G.-G. and I.R.F.; data curation, R.A., S.M., G.C. and A.B.; writing—original draft preparation, R.A., S.M., G.C. and A.B.; writing—review and editing, R.A., S.M., G.C. and A.B. All authors have read and agreed to the published version of the manuscript.

**Funding:** This research received funding from Superstripes onlus. Supported by the National Natural Science Foundation of China (Grant No. 12075273), and work at Stanford University was supported by the U.S. Department of Energy, Office of Basic Sciences, under Contract no. DE-AC02-76SF0051.

**Institutional Review Board Statement:** Not applicable.

**Informed Consent Statement:** Not applicable.

**Data Availability Statement:** The data that support the findings of this study are available from the corresponding author (G.C.) upon reasonable request.

**Acknowledgments:** We thank T. H. Geballe, who passed away during the course of this study, for his long-standing contributions to the study of complexity of superconducting materials, and for his specific contributions to this particular project over a span of many years. We are grateful to Sakura Pascarelli, Olivier Mathon, the ID24 beamline staff and to Michael di Gioacchino for their experimental help. A.A.I. and A.P.M. acknowledge the support of the Ministry of Science and Higher Education of the Russian Federation (Agreement no. 075-15-2021-1352).

**Conflicts of Interest:** The authors declare no conflict of interest. The funders had no role in the design of the study; in the collection, analyses, or interpretation of data; in the writing of the manuscript; or in the decision to publish the results.

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
