# Peer review of "Tensile Microstrain Fluctuations in the BaPbO Units in Superconducting BaPb1−xBixO3 by Scanning Dispersive Micro-XANES"

_condensedmatter, doi:10.3390/condmat8030057_

Round 1

Reviewer 1 Report

In this work, Ruben Albertini et. al analyzed the strain distribution in BaPb1-xBixO3 thin film by inspecting the micro-XANES spectrum at Pb L-edge using synchrotron-based scanning microprobe. Despite of the interesting results, there are remaining issues need to get clarified before consideration for publishing.

1. Details in simulating the XANES spectrum (shown in Figure 2b) are required. Specifically, when using the method of multiple scatting, what are the atomic potential used? What is the package used in the collection, and related critical parameters? A review of the method used should be included, e.g., what is the typical accuracy of the calculation. Reference related to the theoretical background for multiple scattering calucation are missing. 

2. Related to question 1, when comparing the simulated spectrum vs. experiment data (Fig. 3b), could the experimental spectra be directly fitted using the simulated the spectra to derive the strain? Notice that in the simulated spectra, different strain induces energy shift for the white line, and it also alters the shape of the spectrum before the white line. In Fig. 3b, the experimental data does not show visible change in terms of the spectrum shape. Additional comments are highly suggested.

3. In Fig. 4, strain distributions are mapped out for different Pb composition. The composition with Bi=0.19 displays significant variation on the strain variation. Spectrum extracted from different region of interest (ROI) is strongly suggested to be presented and analyzed to validate the results. Especially, what is the noise level of the pixel-wised spectrum? How does noise affect the analysis accuracy?

4. In Fig. 5a, EXAFS of Pb is also presented. Could the author derive the strain (e.g, bond length) from EXAFS and compare with the results derived from XANES analysis?

Author Response

  1. Details in simulating the XANES spectrum (shown in Figure 2b) are required. Specifically, when using the method of multiple scatting, what are the atomic potential used? What is the package used in the collection, and related critical parameters? A review of the method used should be included, e.g., what is the typical accuracy of the calculation. Reference related to the theoretical background for multiple scattering calucation are missing. 

Reply:

We added a sentence in the methods section:

Theoretical simulations of XANES spectra were performed in the framework of full multiple scattering theory for a finite atomic cluster centered at the absorbing Pb atom using the muffin-tin approximation [60-63] and self-consistent field using the implemented FEFF9.6 code as described in reference [64]. The Hedin-Lundqvist exchange potential were adopted and atomic potential were constructed in self-consistent field calculations. The Full Multiple Scattering (MS) calculations were calculated increasing the atomic cluster radius centered at the Pb absorbing atom. The calculated XANES spectra reproduces the experimental data for clusters with radius larger than 5 , and remains invariant for larger clusters, showing that the experimental XANES spectrum probes via multiple scattering only a nanoscale cluster. The instrumental and core-hole induced broadening were consistently used for all simulated spectra to be comparable with experimental spectra with the reference 

 Rehr, J. J., Kas, J. J., Vila, F. D., Prange, M. P., & Jorissen, K. (2010). Parameter-free calculations of X-ray spectra with FEFF9. Physical Chemistry Chemical Physics, 12(21), 5503-5513.

  1. Related to question 1, when comparing the simulated spectrum vs. experiment data (Fig. 3b), could the experimental spectra be directly fitted using the simulated the spectra to derive the strain? Notice that in the simulated spectra, different strain induces energy shift for the white line, and it also alters the shape of the spectrum before the white line. In Fig. 3b, the experimental data does not show visible change in terms of the spectrum shape. Additional comments are highly suggested.

Reply:

We have simulated first the XANES spectra of the original x=0 BaPbO3 phase. The shape of the XANES spectrum is highly dependent on geometry and bond distance of the local cluster of atoms. The strain in the local cluster probed by XANES in superconductive samples with x= 0.19 and 0.28 has been obtained for the doped BPBO simulations by calculations of the energy shift-strain relation due the tensile strain probed by undoped  the expansion of the lattice parameter of the nanoscale cluster relative to the original local lattice parameter of the cluster in BaPbO3.

  1. In Fig. 4, strain distributions are mapped out for different Pb composition. The composition with Bi=0.19 displays significant variation on the strain variation. Spectrum extracted from different region of interest (ROI) is strongly suggested to be presented and analyzed to validate the results. Especially, what is the noise level of the pixel-wised spectrum? How does noise affect the analysis accuracy?

Reply:

The noise level can be evaluated by the width of the strain in the undoped BaPbO3 sample shown in the paper

  1. In Fig. 5a, the EXAFS at the Pb edge is presented. Could the author derive the strain (e.g, bond length) from EXAFS and compare with the results derived from XANES analysis?

Reply:

The limited energy range of the EXAFS spectra and the limited time for data collection of the EXAFS data with the very high signal-to-noise ratio to detect the strain do not allow to make scanning  EXAFS analysis of the micro-domains.

Reviewer 2 Report

The manuscript presents a clear and interesting message regarding the effects of Bi doping on the Pb-O bonds of superconducting perovskites. While I acknowledge the relevance and quality of the work, I would like to ask the authors to clarify a few points before accepting the paper for submission:

  1. All measurements were conducted at room temperature. Could the authors provide their arguments supporting the assumption that the strain distribution will remain consistent during the superconducting phase?

  2. I have a technical question: Since all measurements were performed in a dispersive beamline, it is assumed that the reference was not measured simultaneously. Therefore, I would like to know how the authors ensured energy stability throughout the measurements. In other words, is it possible that the observed energy shift at the Pb edge originated from artifacts within the beamline? Were these maps repeated, or is there only a single measurement?

  3. Regarding sample preparation, what was the thickness of the single crystal used in this study? How did the authors achieve this specific thickness? Could the process of reducing the sample thickness induce any strain in the sample? If not, were the superconducting transitions measured after the single crystal had reached the exact dimensions at which the XAS measurements were performed?"

Author Response

  1. All measurements were conducted at room temperature. Could the authors provide their arguments supporting the assumption that the strain distribution will remain consistent during the superconducting phase?

Reply:

We thank the referee for this comment in fact the measurement temperature was missing in the text. We have added a sentence in the revised version. The reported XANES data were collected at low temperature 50±10 K. The investigation of the local lattice strain at the superconducting phase transition needs more research work planned for future experiments

  1. I have a technical question: Since all measurements were performed in a dispersive beamline, it is assumed that the reference was not measured simultaneously. Therefore, I would like to know how the authors ensured energy stability throughout the measurements. In other words, is it possible that the observed energy shifts at the Pb edge originated from artifacts within the beamline? Were these maps repeated, or is there only a single measurement?

Reply:

During the scanning data collection only the sample moves therefore the measured energy differences between the Multiple scattering resonance and Pb edgecannot be originated from artifacts within the beamline. as tested in undoped samples.

  1. Regarding sample preparation, what was the thickness of the single crystal used in this study? How did the authors achieve this specific thickness? Could the process of reducing the sample thickness induce any strain in the sample? If not, were the superconducting transitions measured after the single crystal had reached the exact dimensions at which the XAS measurements were performed?"

Reply:

The samples have been grown and the homogenous thickness of the crystal for optimal absorption jump edge was reached by standard material science control

Reviewer 3 Report

I am not very familiar with the journal Condensed Matter, which makes it difficult for me to give a clear recommendation on whether or not to publish. I find this manuscript to be difficult to follow, and I question the validity of its results. The data on local strain may be of interest to specialists in scanning X-ray absorption, but this manuscript will not be of broad interest.

The authors propose that local strains due to intrinsic sample inhomogeneity may be key to the superconductivity. The main topic of this paper is a scan of the local strains using an X-ray technique. Two results are the scale of local strain variation, and the fact that the local strain is not correlated to bismuth content, which points to local strain inhomogeneity as an intrinsic effect.

The authors describe the microstrain as originating from lattice mismatch between units of the compound of different compositions. That is a mechanism that should generate large strains on a nanometer scale, but the authors' data are on a micrometer scale. There is also very little explanation of exactly how the local strain is measured – what is the signal and how exactly does it emerge? Fig. 4 shows scans of local microstrain, and strain is given as a scalar quantity. But strain is a tensor, and the authors should define precisely what strain they are looking at. In Fig. 5d, I do not understand how the average microstrain can be zero. The average microstrain is surely just the average strain in the entire sample, and if the sample is not placed under stress that should be zero.

The statement at the end of the Discussion that there is nanoscale phase separation is not supported by the data, which are on a micro- not a nano-meter length scale. The statement in the final sentence, that the high-temperature superconductivity in this compound emerges from the interface between regions with x>0.72 and x<0.72, seems far removed from the dopings studied in this compound: 0.19 and 0.28, and also the doping range over which superconductivity appears. This statement is rather fantastic and is only superficially related to the data in this paper.

Author Response

  1. The authors describe the microstrain as originating from lattice mismatch between units of the compound of different compositions. That is a mechanism that should generate large strains on a nanometer scale, but the authors' data are on a micrometer scale. There is also very little explanation of exactly how the local strain is measured – what is the signal and how exactly does it emerge?

Reply:

The X-ray spot dimension is on a micrometer scale, and the XANES spectroscopy is sensitive to variations of the micriostrain in the probes local 0.5 nm nanocluster in the hundreds of picometers range.  The measured miscrostain in the each probed micron size spot result from statistical distribution of the XANES microstrain-domains. The experimental advances in the field in future will be related to advances on focusing the X.ray beam.

  1. 4 shows scans of local microstrain, and strain is given as a scalar quantity. But strain is a tensor, and the authors should define precisely what strain they are looking at. In Fig. 5d, I do not understand how the average microstrain can be zero. The average microstrain is surely just the average strain in the entire sample, and if the sample is not placed under stress that should be zero.

Reply:

In our study the microstrain value is associated to an energy shift of the XANES multiple scattering resoance. The energy shift of the MSR probes the tensile XANES microstrain  of the 0.5 nm cluster centered at the absorbing Pb atom in BaPb1-xBixO3 irelative to  local nanocluster in the BaPbO3 phase therfore the XANES microstrain is a scalar which changes as the Bismuth content increase.

  1. The statement at the end of the Discussion that there is nanoscale phase separation is not supported by the data, which are on a micro- not a nano-meter length scale. The statement in the final sentence, that the high-temperature superconductivity in this compound emerges from the interface between regions with x>0.72 and x<0.72, seems far removed from the doping studied in this compound: 0.19 and 0.28, and also the doping range over which superconductivity appears. This statement is rather fantastic and is only superficially related to the data in this paper.

Reply:

At the end of the discussion we present our personal, called “fantastic,  ideas on the relevance of this work as a contribution in the hot topic of the role of intrinsic nanoscale phase separation in quantum material which is object of discussions in the main international conferences this year. Clearly these ideas which we like to share with the reader are based also in previous complementary related works [35-39, 73,74]  which could be of interest for the future research work of the reader looking fantastic scientific ideas to be verified by further experimental works.

Round 2

Reviewer 1 Report

The author have not addressed my question #3. To validate the strain/stress analysis from XNAES spectrum, especially at pixel level, please include few XANES spectra extracted from a few region of interests derived from Fig. 4a. 

Author Response

We thank the referee for the constructive comments so we have added the requested figure 6 in the MATERIALS AND METHODS SECTION with the text written in red color    

"In order to establish the actual limitation of the measure of local cluster microstrain by scanning micro XANES in dispersive mode we present in Fig. 6 the typical XANES spectra collected in 12 spots of the 1681 XANES spots collected in the area of 41x41 spots of the pure BaPbO3 sample shown in Fig. 4. The energy of the multiple scattering resonance (EMSR) is given by the difference between the Fermi level at the maximum of first derivative peak and the zero of the XANES spectra derivative around 32.8 eV. The energy fluctuations of the multiple scattering resonance (EMSR) due to the noise level correspond to a noise fluctuation of the microstrain of the order of 0.12% in the undoped sample. Therefore we can clearly measure microstrain spatial fluctuations larger than 0.12% in doped superconducting samples."

Reviewer 2 Report

I am happy with the improvements made by the authors and therefore recommend this manuscript for publication.

Author Response

We thank the referee, to improve the article we have also added a part in the main text written in red color:    

"In order to establish the actual limitation of the measure of local cluster microstrain by scanning micro XANES in dispersive mode we present in Fig. 6 the typical XANES spectra collected in 12 spots of the 1681 XANES spots collected in the area of 41x41 spots of the pure BaPbO3 sample shown in Fig. 4. The energy of the multiple scattering resonance (EMSR) is given by the difference between the Fermi level at the maximum of first derivative peak and the zero of the XANES spectra derivative around 32.8 eV. The energy fluctuations of the multiple scattering resonance (EMSR) due to the noise level correspond to a noise fluctuation of the microstrain of the order of 0.12% in the undoped sample. Therefore we can clearly measure microstrain spatial fluctuations larger than 0.12% in doped superconducting samples."

Reviewer 3 Report

In my first review I misunderstood the authors' method. Although the X-ray beam spot is on the micron scale, the location of a multiple-scattering resonance is correlated with the local strain in a lattice unit.

Nevertheless, I do not find the results of this manuscript to be clear. Strain is not a scalar quantity, and it is unclear what precise lattice distortion the authors observe. The authors observe the microstrain to change by about 1% with Bi substitution. Is that consistent with the change in average lattice constant? 

The main point of this manuscript is that substitution induces substantial strain inhomogeneity. That point does make sense, but the new information provided by the authors is not specific or systematic enough to take this idea further. This work may be of interest for a narrow community interested in structural studies of inhomogeneous materials.

I recommend at this point that the editors decide whether to accept or reject this manuscript. I think that major revision would be required to make it useful to a broad community, and I would prefer not to review it again.

Author Response

(The authors gave the same response as above.)
